# Mean Platelet Volume Predicts Vascular Access Events in Hemodialysis Patients

**DOI:** 10.3390/jcm8050608

**Published:** 2019-05-04

**Authors:** Guillaume Lano, Marion Sallée, Marion Pelletier, Stanislas Bataille, Megan Fraisse, Yaël Berda-Haddad, Philippe Brunet, Stéphane Burtey

**Affiliations:** 1Centre de Néphrologie et Transplantation Rénale, Hôpital de la conception AP-HM, 13005 Marseille, France; guillaume.lano@ap-hm.fr (G.L.); marion.sallee@ap-hm.fr (M.S.); marion.pelletier@ap-hm.fr (M.P.); megan.fraisse@ap-hm.fr (M.F.); philippe.brunet@ap-hm.fr (P.B.); 2Faculty of Pharmacy, Aix Marseille Univ, INSERM, INRA, C2VN, 13005 Marseille, France; stanislas.bataille@sfr.fr (S.B.); 3Elsan, Phocean Institute of Nephrology, Clinique Bouchard, 13008 Marseille, France; 4Hematology and Vascular Biology Department, Hôpital de la Conception, AP-HM, 13005 Marseille, France; yael.berda@ap-hm.fr (Y.B.-H.)

**Keywords:** arteriovenous fistula, hemodialysis, mean platelet volume, thrombosis, end-stage renal disease

## Abstract

Arteriovenous fistula (AVF) and arteriovenous graft (AVG) is the vascular access (VA) of 78% of hemodialysis patients (HD) in France. VA dysfunction corresponding to either stenosis requiring angioplasty or acute thrombosis is responsible for 30% of hospitalizations. Mean platelet volume (MPV) is a biological marker of cardiovascular events. We studied MPV in a cohort of HD patients as a predictive marker of VA dysfunction. We conducted a prospective monocentric cohort study that included patients with AVF or AVG on chronic HD (*n* = 153). The primary outcome was the incidence of VA dysfunction regarding MPV value. The median MPV was 10.8 fL (7.8–13.5), and four groups were designed according to MPV quartiles. Fifty-four patients experienced the first event of VA dysfunction. The incidence of VA dysfunction was higher in patients with the highest MPV: 59% (23 events), 34% (14 events), 27% (11 events), and 18% (6 events), respectively, for the fourth, third, second, and first quartiles (*p* = 0.001). Multivariate analysis confirmed an independent association between MPV and VA dysfunction—OR 1.52 (1.13–2.07), *p* < 0.001. VA dysfunction is predicted by MPV level. Patients with the highest MPV have the highest risk of VA events.

## 1. Introduction

Arteriovenous fistula (AVF) and arteriovenous graft (AVG) is the vascular access (VA) of 78% of hemodialysis (HD) patients in France [1]. Their dysfunction (stenosis and/or thrombosis) is a common complication. VA dysfunction accounts for around 30% of hospitalizations of HD patients [2]. VA dysfunction can lead to VA failure. Thrombosis of VA is enhanced by vessel abnormalities such as neointimal hyperplasia leading to stenosis, endothelial dysfunction, and the procoagulable state observed in patients with chronic kidney disease (CKD).

VA is a non-physiological shunt of the venous vasculature. Vascular anastomosis induces turbulent blood flow and increases shear stress in the venous portion. These abnormalities lead to intimal venous hyperplasia and potential significant stenosis [3]. VA stenosis reduces the arterial inflow and/or increases the outflow venous pressure leading to blood recirculation. VA stenosis is responsive to a decrease in dialysis quality.

Hemostasis disorders are frequent in the course of CKD [4]. Declined glomerular filtration rate increases hemorrhagic risk [4]. Similarly, CKD is associated with cardiovascular [5,6] and venous [7] thrombotic events. Several actors are involved: the acquired uremic thrombopathy described as a pro or antiaggregant [8]; oxidative stress and chronic inflammation; and endothelial dysfunction [9,10]. In the uremic context, several circulating factors have been identified to promote or be associated with VA thrombosis: uremic toxins [11], growth factors such as platelet-derived growth factor (PDGF) and transforming growth factor-beta (TGF-β) [12], antiphospholipid antibodies [13], antithrombin, protein C and S deficiencies, factor V Leiden mutation, and Nitric Oxyde (NO) synthase or plasminogen activator inhibitor-1 abnormalities [4]. Female gender, a history of thrombosis, and unprovoked venous and arterial thrombosis with suspicion of thrombophilia predispose patients to VA stenosis or thrombosis [14,15]. 

To prevent VA thrombosis, close monitoring of clinical and dialysis parameters with blood flow measurements are proposed. Clinical monitoring of VA refers to clinical examination and the ability to puncture the VA with two needles. Dialysis parameters are pre-pump arterial and post-pump venous pressure, Kt/V, and recirculation. Access blood flow and recirculation can be measured via Transonic^®^ during dialysis sessions or by Doppler ultrasound. Unfortunately, these methods cannot be performed at each session and can misdiagnose stenosis.

Platelets have a critical role in primary hemostasis. Platelet function can be approached by mean platelet volume (MPV). MPV is a marker of platelet activity [16]. In acute peripheral thrombocytopenia, MPV increases due to the production of young platelets associated with activated megakaryopoiesis [17]. In chronic thrombocytopenia, higher MPV can be associated with constitutional thrombopathy. MPV is a routine parameter measured or calculated by hematology analyzers in laboratories. A normal value is between 7.5 and 11 fL. MPV is inversely correlated to platelet age [18]. MPV is inversely correlated with platelet counts in order to maintain the total platelet mass constant in one organism [19]. Our aim is to prospectively observe whether MPV can predict VA dysfunction in HD patients.

## 2. Methods

We conducted a single-center prospective observational study at the “hôpital Conception” (Assistance Publique des hôpitaux de Marseille APHM). All hemodialysis patients were screened in the first mid-session of June 2014 (the 4th or 5th of June 2014). Inclusion criteria were an HD patient with vascular access defined as AVF or AVG and HD for more than 3 months. Exclusion criteria were missing laboratory data (platelet count, MPV), the presence of platelet clusters, thrombocytopenia < 50 G/L, refusal to participate, and inability to give informed consent.

Patients were followed prospectively until 1 June 2016 (2 years of follow-up).

Clinical and biological features, comorbidities, treatments, and HD parameters were collected. The population was divided into four subgroups determined by quartiles of MPV: quartile 1 < 25% percentile; quartile 2 between the 25% percentile and the median; quartile 3 between the median and the 75% percentile; quartile 4 ≥ 75% percentile. The evolution of MPV was not followed after inclusion.

A VA event was defined as a composite of the first occurrence of VA thrombosis (no thrill or pulse by physical examination) or hemodynamically significant stenosis requiring endovascular treatment. Significant stenosis was defined during angiography by a reduction of more than 50% [20] of the lumen of the vessel and the need to perform angioplasty. Angiography was performed because of suspicion of VA stenosis with the following:Clinical parameters as extension of compression time, decreasing thrill, collateral venous circulation, edema of the arm with the VA or palpation of a stenosis;and/or dialysis parameters as a drop in KT/v, significant increase in venous pressure, impossibility to obtain the usual VA blood flow;and/or Transonic^®^ or Doppler ultrasound parameters as a decrease in the VA blood flow higher than 20%.

No specific therapeutic intervention was established after angiography or thrombosis apart from monthly flow monitoring of the VA.

VA access was monitored clinically at each HD session (examination and dialysis parameters). Blood flow measurement was performed monthly in all patients who had a history of thrombosis and/or angioplasty on their current VA (high-risk patients).

A VA event was reported for the first of any event occurring during the follow-up period.

Biological assessments were performed at the inclusion in June 2014. Blood samples were drawn at the beginning of the second HD session of the week (Wednesday for Monday–Wednesday–Friday regime session or Thursday for Tuesday–Thursday–Saturday regime session) on an arterial needle into EDTA tubes and transported to the laboratory within 2 h to obtain a full blood count. Platelet and MPV values were obtained using an XN-10 analyzer (Sysmex Corporation™, Kobe, Japan). Platelets were counted using an impedance method with a hydrodynamic focusing system in a fixed volume. Plateletocrit was defined using the cumulative pulse method and MPV was calculated by the ratio plateletocrit/platelets.

The primary outcome was the incidence of VA events. Secondary outcomes were the incidence of each of the composite components: VA thrombosis and VA stenosis.

Data are expressed as the mean and SD for data with normal distribution or median (minimum, maximum) for non-normally distributed data. All numeric variables were tested for normality by the Kolmogorov–Smirnov and Shapiro–Wilk tests. Statistical analyses were performed with the Prism (GraphPad Software Inc., La Jolla, CA, USA) and MedCalc (Ostend, Belgium) software. Differences were considered significant for *p* = 0.05. Baseline variables were compared by chi-square tests for categorical variables, *t*-tests for continuous variables with Gaussian distributions, and Mann–Whitney tests for continuous variables with non-Gaussian distributions. Correlations between MPV and continuous variables were obtained using Spearman correlation coefficients. The Kaplan–Meier method was used to estimate the cumulative VA event-free rate for the time to the first VA events in patients depending on the quartile of MPV. The log-rank test was used to compare the Kaplan–Meier curves. Univariate and multivariate analyses of VA events were performed using a Cox proportional hazard model as a function of MPV as the continuous variable. In multivariate analyses, the model was adjusted for age, sex, platelet count, native AVF, absence of treatment with antiplatelet agents (AAP), antithrombotic treatment, C-reactive protein (CRP), hemoglobin, and treatment with a stimulating agent of erythropoiesis (SAE). Software R [21] and the Survival package were used. The VA event-free survival was censored by death. Only the first event was included in the analysis.

## 3. Results

### 3.1. Baseline Characteristics of the Cohort

Two-hundred and twenty-six (226) hemodialysis patients were screened in the HD center in June 2014. Sixty-four patients were not included (40 patients had a central venous catheter and 24 had begun dialysis sessions in the last three months) and nine patients were excluded (8 for thrombocytopenia and 1 for platelet clusters). The study, therefore, included 153 patients (Figure 1). After a median follow-up of 644 days, 40 patients had died, 20 had received kidney transplantation, and one was lost. Eighty-one patients were still on dialysis in the center and 11 had moved to another dialysis center but medical data were available. Data from patients who underwent a kidney transplantation were recorded after the transplantation. Statistical analysis was performed on the total patient population (153 patients).

The mean MPV was 10.8 fL (7.8–13.5) and the quartiles were as follow: quartile 1: MPV < 10 fL; quartile 2: 10 fL ≤ MPV < 10.7 fL; quartile 3: 10.7 fL ≤ MPV < 11.5 fL; and quartile 4: MPV ≥ 11.5 fL. The characteristics of the population are presented in Table 1. The four quartiles were comparable except for serum creatinine (*p* = 0.05), which was lower in group 1.

### 3.2. Association between MPV and VA Events

Forty-three VA stenosis and 29 VA thrombosis (72) events occurred in 54 patients (35%) with a median time of 259 days. Eighteen patients presented with both events. The first VA event was significant stenosis in 37 patients (69%) and VA thrombosis in 17 patients (31%). Kaplan–Meier analysis (Figure 2) showed a statistically significant difference in the occurrence of VA events depending on the MPV quartile. Twenty-three events (59%) occurred in group 4 (higher quartile), 14 (34%) in group 3, 11 (27%) in group 2, and 6 (18%) in group 1 (*p* = 0.001).

Table 2 presents the demographic and biological characteristics between patients with a VA event during the follow-up and those without a VA event. MPV (11.3 vs. 10.6 fL, *p* < 0.001) and platelet counts (161 vs. 222 G/L, *p* < 0.001) differed between these two groups.

The Cox multivariate analysis (Table 3) showed an independent association between the MPV and the occurrence of any VA event; the odds ratio (OR) was 1.58 (1.17–2.14; *p* = 0.003) for every 1 fL increase of MPV.

### 3.3. Secondary Outcomes

Forty-three patients presented with VA stenosis (28%) with a median time of 268 days. The Kaplan–Meier analysis (Figure 3A) showed a statistical difference in the occurrence of VA stenosis according to MPV quartiles with a higher incidence in the higher MPV group: 19 (49%) patients presented with VA stenosis in group 4, 11 (27%) in group 3, 7 (17%) in group 2, and 6 (18%) in group 1 (*p* = 0.005).

In the Cox model, the OR of VA stenosis was 1.55 (1.10–2.18; *p* = 0.001) for every 1 fL increase of MPV (Table 4). Native versus graft VA (OR at 0.37 (0.13–0.96); *p* = 0.04) is a protective factor.

Twenty-nine patients presented with VA thrombosis (19%) with a median time of 322 days. The Kaplan–Meier analysis (Figure 3B) did not show any difference between the groups. There was a trend (*p* = 0.14) to a higher incidence in group 4 with 11 events (28 %), 7 (17%) in group 3, 8 (20%) in group 2, and 3 (9%) in group 1. The Cox model showed a trend for MPV with OR at 1.32 (0.90–1.92; *p* = 0.16) (Table 5).

## 4. Discussion

MPV is a predictive factor of VA events (significant stenosis and acute thrombosis). The risk of presenting with a VA event is three times higher in patients with the highest MPV values compared to those with the lowest MPV values. We showed a strong association between the increase in MPV and VA stenosis. The Cox model confirmed the protective character of an AVF compared to an AVG in the occurrence of VA stenosis (and a trend for VA events), which is not surprising. MPV could be a biomarker of VA dysfunction and could help clinicians identify high-risk populations.

Platelets play a role in VA stenosis development. The key event in VA failure is intimal hyperplasia. Platelets are involved in intimal hyperplasia, like shear stress, hypoxic injury, inflammation, uremia, and thrombosis [22]. Intimal hyperplasia is defined as the abnormal migration and proliferation of vascular smooth muscle cells in response to an injury of the vessel and is promoted by the inflammatory response. Platelets are found in the intimal hyperplasia of explanted vascular prostheses [19]. Large platelets with high MPV, containing denser α granules, aggregate faster after ADP or collagen stimulation and secrete PDGF, TGF-β [23], fibrinogen, and beta-thromboglobulin [12]. PDGF and TGF-β are two well-known factors implicated in the initiation of intimal hyperplasia. PDGF is a growth factor involved in the development of intimal hyperplasia [24] and stimulates smooth muscle cells and inflammatory cell mobilization, migration, and proliferation. TGF-β is a pro-inflammatory cytokine involved in the development of atheromatous plaques. PDGF [22] and TGF-β [25] are described in a large amount in the intima of VA stenosis. In addition, platelets are important in the initiation of thrombosis and activated platelets could increase the thrombotic risk. Higher platelet aggregability was reported in patients with VA dysfunction [26]. Stenosis promotes thrombosis in VA. In our study, MPV is not associated with VA thrombosis. It is probably due to a low number of events. Thrombosis can be prevented by having a closer look at stenosis. An active screening strategy by systematic Doppler or Transonic^®^ flow rates could reduce the risk of thrombotic events of VA [27,28]. European and international guidelines recommend regular and objective monitoring of access function (ideally every month for AVG and every 3 months for AVF) [29]. A significant decrease in access flow over 20% requires endovascular exploration and, if necessary, stenosis treatment by endovascular or surgical methods [29]. Detection of stenosis by frequent flow measurement and rapid angiography achievement could explain our low thrombosis rate despite the high-risk profiles of the patients.

Interestingly, we confirmed the fact that an increase in MPV is associated with a reduction in platelet counts. For thrombotic risk, platelet activity is more critical than platelet count [16]. MPV is an easy and accessible tool to approach the reactivity of platelets in HD patients. One limitation of our study is that we did not test the influence of the AVF or AVG sites (upper or lower arm) and the cannulation technique (heterogeneous in our center), which are two important factors in stenosis or thrombotic risk [30,31]. Moreover, we did not have information about a possible antecedent of thrombophilia in our cohort.

This study does not allow conclusions about the role of activated platelets in vascular access failure. High MPV could be a marker of dysfunctional VA. VA stenosis could stimulate hemostasis, inducing platelet consumption leading to an increase in their turnover and production of larger platelets. This hypothesis is promoted by the inverse correlation between platelet count and MPV. It would be of interest to study MPV evolution after VA failure.

The threshold of the MPV value depends on the method of measurement used by the hematology analyzer and cannot be generalized. However, centers using an XN-10 could use our value to predict the risk of VA dysfunction.

In the general population, the highest MPV is associated with cardiovascular (CV) events [32,33] and with mortality during CV disease. In CKD, MPV is increased. It is inversely correlated with glomerular filtration rate. Two studies have found a link between high MPV and coronary heart disease [34,35]. A recent study of a national retrospective cohort of 149,000 chronic HD patients found a correlation between an increase in MPV and mortality [36] but not with cardiovascular events. We identified a group of patients at high risk of VA events, those with the highest MPV (≥11.5 fL). Our results are in agreement with two recent studies [37,38]. This population could benefit from more aggressive stenosis monitoring with Transonic© or Doppler ultrasound leading to preventive treatment of thrombosis. It would be interesting to study a larger cohort for the effect of this strategy on the rate of VA longevity and failure. Preventive pharmacologic strategies did not show any improvement in preventing VA thrombosis [39,40] in HD patients. Identification of a biological marker to determine VA thrombosis in high-risk populations could be useful for designing future clinical trials [41]. The role of platelets in AVF thrombosis remains to be demonstrated and could be investigated by platelet reactivity testing.

In conclusion, MPV values predict the risk of VA events in HD patients. MPV is a universal parameter, included in each full blood count without additional cost or delay for clinicians. We suggest testing its efficiency in a larger population. MPV could lead to better care of VA, the line of life for HD patients.

## Figures and Tables

**Figure 1 jcm-08-00608-f001:**
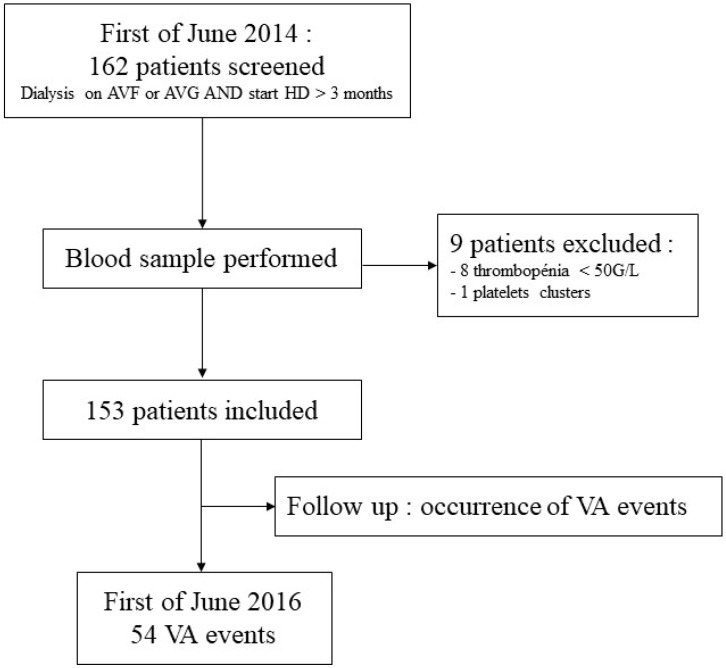
Flow chart 1.

**Figure 2 jcm-08-00608-f002:**
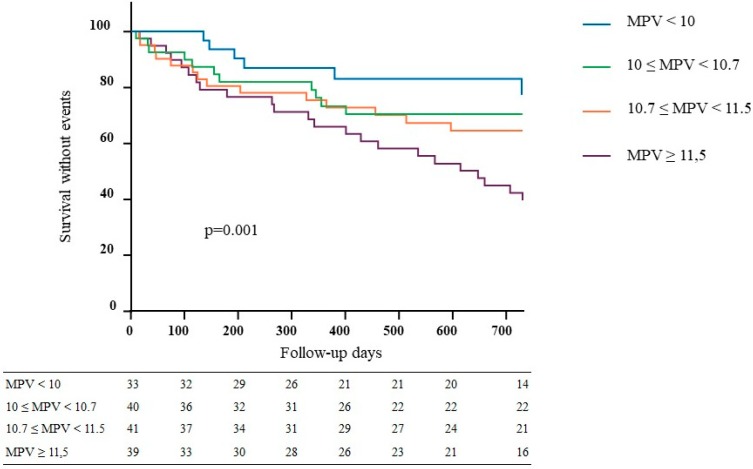
Kaplan–Meier plots of vascular access events: 54 events occurred during the follow-up with group 1: 18% *n* = 6/33 patients; group 2: 27% *n* = 11/40 patients; group 3: 34% *n* = 14/41 patients; and group 4: 59% *n* = 23/39 patients.

**Figure 3 jcm-08-00608-f003:**
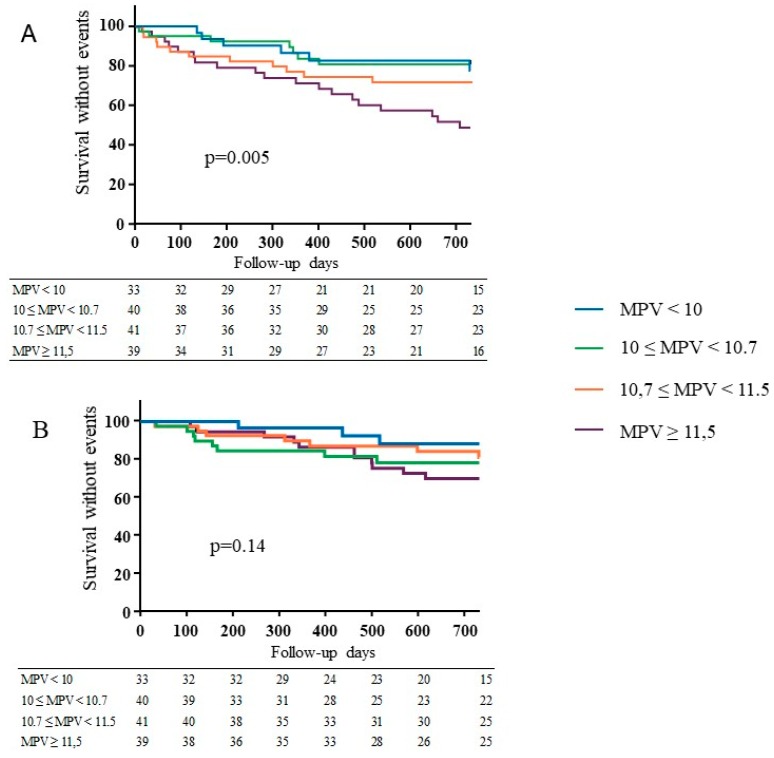
(**A**) Kaplan–Meier plots of vascular access stenosis: 43 events occurred during the follow up with group 1: 18% *n* = 6/33 patients; group 2: 17% *n* = 7/40 patients; group 3: 27% *n* = 11/41 patients; group 4: 49% *n* = 19/39 patients. (**B**) Kaplan–Meier plots of vascular access thrombosis: 29 events occurred during the follow-up with group 1: 9% *n* = 3/29 patients; group 2: 20% *n* = 8/29 patients; group 3: 17% *n* = 7/29 patients; and group 4: 28% *n* = 11/29 patients.

**Table 1 jcm-08-00608-t001:** Baseline characteristics of the cohort.

Patients Characteristics	Quartile 1	Quartile 2	Quartile 3	Quartile 4	All Population	*p*
	*N* = 33	*N* = 40	*N* = 41	*N* = 39	*N* = 153	
Age, years, mean (SD)	68.1 (16.9)	64.8 (15.9)	65.8 (14.5)	63.7 (19)	65.5 (16.5)	0.66
Female (%)	16 (48)	18 (45)	13 (32)	15 (38)	62 (41)	0.46
BMI (kg/m^2^), mean (SD)	23.6 (4.4)	24.8 (4.4)	25.2 (5.2))	24.4 (4.7)	24.6 (4.6)	0.56
ESRD vintage, years. mean (SD)	8.1 (8.4)	6.9 (10.5)	8.5 (8.1)	7.9 (8.7)	7.8 (8.8)	0.44
Transplantation *n* (%)	5 (15)	3 (7)	6 (15)	8 (20)	22 (14)	0.43
Vascular access *n* (%)						
AVF	24 (73)	31 (78)	34 (82)	31 (79)	120 (78)	0.76
AVG	9 (27)	9 (22)	7 (18)	8 (21)	33 (22)	0.76
Comorbidities (%)						
Congestive heart failure	10 (30)	10 (25)	8 (20)	10 (26)	38 (25)	0.45
Atrial fibrillation	9 (27)	12 (30)	11 (27)	14 (36)	46 (30)	0.81
Ischemic heart disease	9 (27)	12 (30)	17 (41)	16 (41)	54 (35)	0.45
Stroke	4 (12)	6 (15)	2 (5)	7 (18)	19 (12)	0.32
Venous thrombosis	8 (24)	4 (10)	4 (10)	3 (8)	21 (14)	0.14
Dyslipidemia	13 (36)	12 (30)	17 (41)	20 (51)	61 (40)	0.26
Tobacco past/present	11 (35)	17 (49)	18 (50)	20 (54)	66 (43)	0.47
Diabetes	10 (30)	15 (37)	15 (37)	15 (38)	55 (40)	0.91
Hypertension	28 (85)	37 (92)	35 (85)	35 (90)	135 (88)	0.69
Peripheral vascular disease	7 (21)	10 (25)	11 (27)	12 (31)	40 (55)	0.83
Cancer	6 (19)	10 (25)	8 (20)	6 (15)	30 (20)	0.76
Laboratory variable mean (± SD)						
Hemoglobin (g/dL)	10.8 (1.9)	10.6 (1.4)	10.6 (1.1)	10.7 (1.0)	10.7 (1.2)	0.9
Platelet count (G/L)	219 (76)	225 (87)	205 (60)	153 (47)	200 (74)	<0.001
MPV (fL)	9.5 (0.5)	10.3 (0.1)	11.0 (0.2)	12.8 (0.5)	10.8 (1.1)	<0.001
Leukocytes (G/L)	6.8 (1.9)	7.4 (2.9)	6.6 (2.7)	6.4 (2.1)	6.7 (2.5)	0.36
Beta2 microglobulin (mg/L)	28.4 (7.8)	26.7 (6.5)	29.4 (5.3)	26.9 (8)	27.9 (6.9)	0.09
Urea (mg/dL)	56.3 (12.9)	61.9 (19.3)	59.1 (21.6)	59.7 (14.0)	59.7 (17.6)	0.81
Creatinine (mg/dL)	7.96 (2.58)	8.87 (2.62)	9.60 (3.00)	8.50 (2.15)	8.78 (2.65)	0.05
Potassium (mmol/L)	4.8 (0.6)	5.1 (0.8)	5.0 (0.7)	4.9 (0.6)	5.0 (0.7)	0.32
Bicarbonate (mmol/L)	20.3 (2.1)	20.6 (2.4)	21.1 (2.5)	21.3 (2.4)	21.0 (2.3)	0.53
Calcium (mg/dL)	9.38 (2.04)	9.58 (0.48)	9.54 (0.68)	9.30 (0.68)	9.46 (0.68)	0.27
Phosphorus (mg/dL)	4.77 (1.58)	5.17 (2.26)	5.20 (2.20)	4.49 (1.52)	4.96 (1.86)	0.34
Albumin (g/L)	36.6 (5.3)	38.2 (4.8)	39.4 (5.0)	38.7 (4.9)	38.3 (5.0)	0.06
Parathyroid hormone (pg/mL)	23.8 (26.9)	29.8 (31.2)	30.1 (39.5)	29.3 (41.2)	28.6 (35.6)	0.9
C reactive protein (mg/L)	19.1 (25.5)	10.8 (15.7)	17.5 (41.7)	10.1 (16.6)	14.2 (27.3)	0.28
Iron saturation (%)	26 (10)	27 (18)	23 (9)	31 (14)	27 (13)	0.06
Ferritin (µg/L)	580 (319)	465 (278)	494 (345)	530 (291)	515 (310)	0.38
Dialysis modality						
Hemodialysis *n* (%)	13 (40)	13 (32)	14 (35)	21 (54)	61 (40)	0.20
Hemofiltration/Hemodiafiltration *n* (%)	20 (60)	27 (67)	27 (65)	18 (46)	92 (60)	0.20
Kt/V. mean (SD)	1.41 (0.25)	1.50 (0.28)	1.53 (0.30)	1.53 (0.21	1.50 (0.26)	0.31
Time per week. hours. mean (SD)	12.3 (1.7)	13.5 (2.6)	13.5 (2.63)	12.8 (2.1)	13.1 (2.2)	0.08
Medication						
ACEI/ARB *n* (%)	5 (17)	11 (27)	9 (22)	13 (33)	38 (25)	0.32
Beta-blocker *n* (%)	10 (30)	19 (47)	18 (44)	20 (51)	67 (44)	0.31
Calcium channel blocker *n* (%)	3 (9)	9 (22)	9 (22)	11 (28)	32 (21)	0.25
Diuretic *n* (%)	10 (30)	14 (35)	18 (43)	16 (41)	58 (38)	0.63
Antiplatelet agent *n* (%)	15 (45)	19 (47)	22 (54)	23 (60)	79 (52)	0.64
Aspirin *n* (%)	11 (35)	20 (56)	20 (53)	14 (42)	65 (42)	0.33
Clopidogrel *n* (%)	4 (12)	8 (20)	9 (22)	11 (28)	32 (21)	0.42
Vitamin K antagonist *n* (%)	6 (18)	7 (17)	6 (15)	12 (31)	31 (20)	0.29
Statin *n* (%)	7 (21)	12 (30)	17 (41)	16 (41)	52 (34)	0.21
Intravenous iron *n* (%)	20 (61)	24 (60)	22 (54)	24 (62)	90 (59)	0.9
Erythropoietin. µg/s (SD)	57 (52)	52 (60)	62 (70)	33 (30)	51 (55)	0.10

Quartile 1. MPV < 10; Quartile 2. 10 ≤ MPV < 10.7; quartile 3. 10.7 ≤ MPV < 11.5; quartile 4. MPV ≥ 11.5. MPV in fL. Abbreviations: MPV: mean platelets volume, BMI: Body Mass Index, ESRD: End-stage Renal Disease; VA: vascular access; AVF: arteriovenous fistula; AVG: arteriovenous graft; ACEI: angiotensin-converting-enzyme inhibitor; ARB: angiotensin II receptor blockers; SD: Standard deviation. Note: Conversion factors for units: serum creatinine in mg/dL to μmol/L: ×88.4; serum urea in mg/dL to mmol/L: ×0.357; serum calcium in mg/dL to mmol/L: ×0.2495; serum phosphorus mg/dL to mmol/L: ×0.3229; serum vitamin D (25-OH vit D) in ng/mL to nmol/mL: ×2.4.

**Table 2 jcm-08-00608-t002:** Baseline characteristics of patient with vascular access events or not.

Patients Characteristics	With VA Events	Without VA Events	*p*
	*n* = 54	*n* = 99	
Age, years, mean (SD)	66.0 (18.1)	65.3 (15.7)	0.62
Female (%)	21 (39)	41 (41)	0.86
BMI (kg/m^2^), mean (SD)	24.5 (4.6)	24.6 (4.6)	0.88
ESRD vintage, years, mean (SD)	8.4 (9.0)	7.5 (8.5)	0.27
Transplantation *n* (%)	8 (15)	14 (14)	0.43
Vascular access *n* (%)			
AVF	38 (70)	82 (83)	0.10
AVG	16 (30)	17 (17)	0.10
Comorbidities (%)			
Congestive heart failure	16 (30)	22 (22)	0.33
Atrial fibrillation	17 (31)	29 (29)	0.86
Ischemic heart disease	19 (35)	35 (35)	0.90
Stroke	10 (19)	9 (9)	0.12
Venous thrombosis	8 (15)	13 (13)	0.52
Dyslipidemia	19 (35)	42 (42)	0.40
Tobacco past/present	22 (41)	44 (44)	*0.60*
Diabetes	18 (33)	37 (37)	0.72
Hypertension	48 (89)	87 (88)	0.90
Peripheral vascular disease	16 (30)	24 (24)	0.56
Cancer	9 (17)	21 (21)	0.52
Laboratory variable mean (± SD)			
Hemoglobin (g/dL)	10.7 (1.4)	10.6 (1.2)	0.68
Platelet count (G/L)	161 (56)	222 (74)	<0.001
MPV (fL)	11.3 (1.2)	10.6 (0.9)	<0.001
Leukocytes (G/L)	6.3 (0.3)	6.9 (2.7)	0.23
Beta2 microglobulin (mg/L)	27.7 (7.2)	27.9 (6.7)	0.9
Urea (mg/dL)	58.0 (22.4)	60.8 (14.0)	0.08
Creatinine (mg/dL)	8.76 (2.93)	8.78 (2.51)	0.84
Potassium (mmol/L)	5.0 (0.8)	5.0 (0.6)	0.87
Bicarbonate(mmol/L)	21.2 (2.6)	20.9 (2.2)	0.29
Calcium (mg/dL)	9.46 (0.64)	9.48 (1.08)	0.73
Phosphorus (mg/dL)	4.86 (1.24)	4.96 (1.86)	0.36
Albumin (g/L)	38.4 (5.5)	38.2 (4.8)	0.57
Parathyroid hormone (pg/mL)	31 (40)	27 (33)	0.62
C reactive protein (mg/L)	11.5 (17.6)	15.7 (31.3)	0.61
Iron saturation (%)	28 (14)	27 (13)	0.69
Ferritin (µg/L)	488 (285)	530 (323)	0.69
Dialysis modality			
Hemodialysis *n* (%)	25 (46)	36 (36)	0.30
Hemofiltration/Hemodiafiltration *n* (%)	29 (54)	63 (64)	0.30
Kt/V, mean (SD)	1.51 (0.31)	1.50 (0.23)	0.9
Time per week, hours, mean (SD)	13.2 (2.2)	13 (2.3)	0.50
Medication			
ACEI/ARB *n* (%)	16 (30)	23 (23)	0.43
Beta-blocker *n* (%)	27 (50)	41 (41)	0.39
Calcium channel blocker *n* (%)	13 (24)	18 (18)	0.40
Diuretic *n* (%)	14 (26)	43 (43)	0.39
Ant platelet agent *n* (%)	26 (48)	53 (52)	0.61
Aspirin *n* (%)	21 (39)	45 (45)	0.50
Clopidogrel *n* (%)	14 (26)	15 (15)	0.13
Vitamin K antagonist (%)	10 (19)	21 (21)	0.67
Statin *n* (%)	22 (41)	30 (30)	0.28
Intra venous iron *n* (%)	30 (56)	60 (61)	0.49
Erythropoietin, µg/s (SD)	50 (56)	51 (58)	0.45

Note: Conversion factors for units: serum creatinine in mg/dL to μmol/L: ×88.4; serum urea in mg/dL to mmol/L: ×0.357; serum calcium in mg/dL to mmol/L: ×0.2495; serum phosphorus mg/dL to mmol/L: ×0.3229; serum vitamin D (25 OH vit D) in ng/mL to nmol/mL: ×2.4.

**Table 3 jcm-08-00608-t003:** Multivariate analysis by the Cox model of vascular access events.

	Odds Ratios (95% CI) of VA Events	*p*
Age/years	0,99 (0.97–1)	0.09
Sex (female)	1.42 (0.77–2.60)	0.26
MPV	1.58 (1.17–2.14)	0.003
platelets	1 (0.99–1)	0.7
AVF vs. AVG	0.48 (0.22–1.08)	0.08
No APA	0.72 (0.38–1.36)	0.31
No AVK	1.07 (0.52–2.18)	0.86
CRP	1 (0.99–1.01)	0.20
Hemoglobin	0.83 (0.59–1.17)	0.29
ESA	0.99 (0.99–1)	0.40

Note: Cox model with time-dependent variables. The covariates used were chosen from the variables described in the literature as having an impact on the functionality of the VA and/or significant in univariate model. Abbreviations: MPV, mean platelets volume; VA, vascular access; AVF, arterio-venous fistula; AVG, arterio-venous graft; APA, anti-platelets agents; AVK, anti-vitamin K; CRP, C-reactive-protein; ESA, Erythropoiesis Stimulating Agent.

**Table 4 jcm-08-00608-t004:** Multivariate analysis by the Cox model of vascular access stenosis.

	Odds Ratios (95% CI) of VA Stenosis	*p*
Age/years	0.99 (0.97–1)	0.13
Sex (female)	1.78 (0.89–3.53)	0.10
MPV	1.55 (1.10–2.18)	0.001
platelets	1 (0.99–1)	0.95
AVF vs. AVG	0.37 (0.13–0,96)	0.04
No APA	0.70 (0.33–1.38)	0.28
No AVK	1.20 (0.55–2.65)	0.66
CRP	1 (1–1.02)	0.03
Hemoglobin	0.84 (0.57–1.25)	0.39
ESA	0.99 (0.98–1.00)	0.33

Note: Cox model with time-dependent variables. The covariates used were chosen from the variables described in the literature as having an impact on the functionality of the VA and/or significant in univariate model.

**Table 5 jcm-08-00608-t005:** Multivariate analysis by cox model of vascular access thrombosis.

	Odds Ratios (95% CI) of VA Thrombosis	*p*
Age/years	0.99 (0.97–1.02)	0.70
Sex (female)	0.75 (0.32–1.77)	0.51
MPV	1.32 (0.90–1.92)	0.16
platelets	1 (0.99–1)	0.80
AVF vs. AVG	0.83 (0.31–2.21)	0.71
No APA	0.50 (0.21–1.19)	0.12
No AVK	1.07 (0.40–2.83)	0.89
CRP	1 (0.99–1.02)	0.26
Hemoglobin	0.83 (0.53–1.29)	0.41
ESA	1 (0.99–1.00)	0.70

Note: Cox model with time-dependent variables. The covariates used were chosen from the variables described in the literature as having an impact on the functionality of the VA and/or significant in univariate model. Abbreviations: MPV, mean platelets volume; VA, vascular access; AVF, arterio-venous fistula; AVG, arterio-venous graft; APA, anti-platelets agents; AVK, anti-vitamin K; CRP, C-reactive-protein; ESA, Erythropoiesis Stimulating Agent.

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
