# Peer review of "Mean Platelet Volume Predicts Vascular Access Events in Hemodialysis Patients"

_jcm, 2019, doi:10.3390/jcm8050608_

Reviewer 1 Report

The authors suggest the MPV as a novel marker for vascular access at risk. The statistical methods they used appear appropriate. However, as the current guidelines on vascular access care suggest use the access flow and its changes for this, which is more laborious than a simple and cheap MPV test, it would be interesting to compare sensitivity and specificity of both.

As part of the investigated patient cohort apparently underwent some treatment of the stenoses, it would be interesting to include the information whether the MPV decreased after such treatment (if successful) as a sign of improved access condition.

Minor comments:

The statement on lines 54-55 (p. 2) is rather confusing suggesting that the Transonic device uses thermodilution to measure access blood flow and recirculation while it is based on ultrasonic dilution (thermodilution is used in the Fresenius BTM module for recirculation measurement only).

Additionally – Kt/V cannot be used to detect stenosis if it is between the needles.

Which two quartiles are compared by the p-value given in the right column of the Tab. 1?

Tab. 1 can be significantly shortened or abbreviated. Its current form is unnecessarily extensive.

The abbreviation AAP on line112, p. 3 should be explained before being used.

English of the article should be checked (e.g. line 86-87 should be a decrease in VA blood flow higher than 20%” instead of upper 20%; Bbloquant in Tab. 1, p.6 should be Beta-blocker etc.).

Author Response

Answers (in bold) to reviewer 1

The authors suggest the MPV as a novel marker for vascular access at risk. The statistical methods they used appear appropriate. However, as the current guidelines on vascular access care suggest use the access flow and its changes for this, which is more laborious than a simple and cheap MPV test, it would be interesting to compare sensitivity and specificity of both.

As part of the investigated patient cohort apparently underwent some treatment of the stenoses, it would be interesting to include the information whether the MPV decreased after such treatment (if successful) as a sign of improved access condition.

We thank the reviewer for this very interesting comment. Unfortunately, in the limited time for the review we are not able to check if MPV decreased after the treatment of the stenosis. We planned to check it for a new work.

Minor comments:

-The statement on lines 54-55 (p. 2) is rather confusing suggesting that the Transonic device uses thermodilution to measure access blood flow and recirculation while it is based on ultrasonic dilution (thermodilution is used in the Fresenius BTM module for recirculation measurement only). To avoid any confusion, we removed “with a thermodilution technique”

Additionally – Kt/V cannot be used to detect stenosis if it is between the needles. We agree.

-Which two quartiles are compared by the p-value given in the right column of the Tab. 1?

P value in table 1 compare all the column. Posttest as Bonferroni comparing all pairs of columns were not performed because of the effect in each group.

Tab. 1 can be significantly shortened or abbreviated. Its current form is unnecessarily extensive. We agree with this comment and removed uremic toxins, etiologies of CKD, respiratory disease, neutrophil, lymphocytes count, gammaglobulin, 25OHvitD predialysis blood pressure.

The abbreviation AAP on line 112, p. 3 should be explained before being used. We corrected it.

English of the article should be checked (e.g. line 86-87 should be a decrease in VA blood flow higher than 20%” instead of upper 20%; Bbloquant in Tab. 1, p.6 should be Beta-blocker etc.). We checked and corrected all the typos.

Reviewer 2 Report

This study addresses an important clinical need for a fast and effective method to predict patients undergoing hemodialysis at risk of VA events.  The prospective study collected data including clinical parameters, full blood count, treatments and comorbidities from patients for a 2-year period.  The authors demonstrated that stratifying patients according to their mean platelet volume allowed for predication of increased risk of VA stenosis.  Use of a routinely obtained parameter for prediction of VA event risk has obvious benefits with no additional cost and a short time scale helping to ensure effective life saving hemodyalsis.

Some points for consideration are below

1.     There are some grammatical errors and differences in font size, please amend throughout.

2.     Page 2 Line 93.  What do the authors mean by the beginning of the middle week session?  Was this the case for every blood sample?

3.     Page 2, Line 63 – This sentence is not clear, is it missing something?  Please clarify.

4.     Page 3 Line 120 – 121- repetition ’24 because hemodialysis was started during the last 3 months)’

5.     Were patients still followed up after treatment of a VA event?  Were they put on antiplatelet treatments? Was MPV predicative of subsequent incidences? 

6.     Figure 1 suggests all 226 patients were recruited on the same day.  Please clarify.  Figure 1 does little to add to the message of the manuscript.  It would be better to have more detail on the analysis done on the patients/samples.  For example, clarification on when blood samples are taken, angiography performed etc..

7.     The authors state that the population characteristics were comparable except creatinine level but iron saturation also appears to be significant.  Please clarify.

8.     The authors state that the low rate of thrombosis could be due to aggressive monitoring but it is not clear what the normal protocol is for monitoring HD patients e.g. are blood sample in addition to routine bloods? Were blood samples taken at multiple points or only the initial admittance to the study.

9.     The authors mention that their obtained threshold MPV value could be used to predict risk of VA dysfunction if other centres were using the same analyser but the threshold differs with different analysers.  This would limit the usefulness of the data.  How would they propose to standardise the method for multicentre use? 

10.  Differences in the platelet count were also observed.  Did low platelet count predict VA events independently?

11.  No significant difference in thrombotic risk was observed.  It would be useful to study the relationship of MPV with platelet reactivity, (platelet aggregation, P-selectin and phosphatidylserine exposure), does MPV correlate with changes in platelet function in this group.  

Author Response

Answers (in bold) Reviewer 2

This study addresses an important clinical need for a fast and effective method to predict patients undergoing hemodialysis at risk of VA events.  The prospective study collected data including clinical parameters, full blood count, treatments and comorbidities from patients for a 2-year period.  The authors demonstrated that stratifying patients according to their mean platelet volume allowed for predication of increased risk of VA stenosis.  Use of a routinely obtained parameter for prediction of VA event risk has obvious benefits with no additional cost and a short time scale helping to ensure effective life saving hemodyalsis.

Some points for consideration are below

1.     There are some grammatical errors and differences in font size, please amend throughout. We checked and corrected it

2.     Page 2 Line 93.  What do the authors mean by the beginning of the middle week session?  Was this the case for every blood sample? We corrected it “Every blood samples were drawn at the beginning of the second HD sessions of the week (Wednesday for Monday-Wednesday-Friday regime session or Thursday for Tuesday-Thursday-Saturday regime session)”  

3.     Page 2, Line 63 – This sentence is not clear, is it missing something?  Please clarify.  The new sentence is “MPV is inversely correlated with platelet counts, in order to maintain the total platelet mass constant in one organism”

4.     Page 3 Line 120 – 121- repetition ’24 because hemodialysis was started during the last 3 months)’We removed it.

5.     Were patients still followed up after treatment of a VA event? Yes Were they put on antiplatelet treatments? Not systematically, we add a sentence to clarify it.

Was MPV predicative of subsequent incidences?  We did not check the multiple events. We add a sentence:” Evolution of MPV were not followed after inclusion.”

6.     Figure 1 suggests all 226 patients were recruited on the same day.  Please clarifyThe patients were recruited on two days as stated in this new sentence: “All hemodialysis patients were screened in the first mid-session of June 2014 (the 4th or the 5th of June 2014)”

 Figure 1 does little to add to the message of the manuscript.  It would be better to have more detail on the analysis done on the patients/samples.  For example, clarification on when blood samples are taken, angiography performed etc.. We proposed a new version of figure 1 in the manuscript.

7.     The authors state that the population characteristics were comparable except creatinine level but iron saturation also appears to be significant.  Please clarify. The p-value is in fact 0,056 so non significative. We corrected the p-value to 0.06 in table 1.

8.     The authors state that the low rate of thrombosis could be due to aggressive monitoring but it is not clear what the normal protocol is for monitoring HD patients e.g. are blood sample in addition to routine bloods? High risk patient is patient who had history of thrombosis and/or angioplasty on their current VA. We performed blood flow measurement (transonic or Doppler) monthly for these patients.

 Were blood samples taken at multiple points or only the initial admittance to the study. The MPV quartile is determined only with the initial measurement

9.     The authors mention that their obtained threshold MPV value could be used to predict risk of VA dysfunction if other centres were using the same analyser but the threshold differs with different analysers.  This would limit the usefulness of the data.  How would they propose to standardise the method for multicentre use? Our analyzer is used in more than 2/3 of hematology lab in France, we don’t know the frequency in the Europe and worldwide, so our data could be use directly in a vast majority of centers. For the centers using another analyzer, they could perform the same type of analyze to measure the quartiles of the values of MPV and used it as us.  

10.  Differences in the platelet count were also observed.  Did low platelet count predict VA events independently? In multivariable analysis we observed low platelets number is not associated with the risk of VA events so low platelets count don’t predict VA events.

11.  No significant difference in thrombotic risk was observed.  It would be useful to study the relationship of MPV with platelet reactivity, (platelet aggregation, P-selectin and phosphatidylserine exposure), does MPV correlate with changes in platelet function in this group.    This remark is very interesting and could be the subject of a new work on platelets functionality. All these studies must be performed on fresh blood drawing, so we could not answer for our cohort. We add a sentence in the discussion to state it.